# Improved YOLOv8-Seg Network for Instance Segmentation of Healthy and Diseased Tomato Plants in the Growth Stage

Xiang Yue [ID], Kai Qi, Xinyi Na [ID], Yang Zhang [ID], Yanhua Liu [ID] and Cuihong Liu *

College of Engineering, Shenyang Agricultural University, Shenyang 110866, China; yuexiang@syau.edu.cn (X.Y.); 2022240067@stu.syau.edu.cn (K.Q.); 2022240052@stu.syau.edu.cn (X.N.); 2022240040@stu.syau.edu.cn (Y.Z.); 2022240006@stu.syau.edu.cn (Y.L.)
* Correspondence: cuihongliu77@syau.edu.cn

**Abstract:** The spread of infections and rot are crucial factors in the decrease in tomato production. Accurately segmenting the affected tomatoes in real-time can prevent the spread of illnesses. However, environmental factors and surface features can affect tomato segmentation accuracy. This study suggests an improved YOLOv8s-Seg network to perform real-time and effective segmentation of tomato fruit, surface color, and surface features. The feature fusion capability of the algorithm was improved by replacing the C2f module with the RepBlock module (stacked by RepConv), adding SimConv convolution (using the ReLU function instead of the SiLU function as the activation function) before two upsampling in the feature fusion network, and replacing the remaining conventional convolution with SimConv. The F1 score was 88.7%, which was 1.0%, 2.8%, 0.8%, and 1.1% higher than that of the YOLOv8s-Seg algorithm, YOLOv5s-Seg algorithm, YOLOv7-Seg algorithm, and Mask RCNN algorithm, respectively. Meanwhile, the segment mean average precision (segment mAP$_{@0.5}$) was 92.2%, which was 2.4%, 3.2%, 1.8%, and 0.7% higher than that of the YOLOv8s-Seg algorithm, YOLOv5s-Seg algorithm, YOLOv7-Seg algorithm, and Mask RCNN algorithm. The algorithm can perform real-time instance segmentation of tomatoes with an inference time of 3.5 ms. This approach provides technical support for tomato health monitoring and intelligent harvesting.

**Keywords:** YOLOv8; instance segmentation; disease detection; maturity segmentation



## 1. Introduction

The accurate segmentation of growing and diseased tomatoes, even against complex backgrounds, is essential for effective tomato picking, fruit monitoring, and precise evaluation of tomato size and quality [1]. Accurate and timely segmentation can benefit various applications, such as machine vision and greenhouse monitoring systems [2]. There are several challenges faced by tomato instance segmentation at present. Environmental factors such as changes in lighting, overlapping fruit, leaf occlusion, and variations in angles can interfere with the process. Moreover, alterations in the surface color and features of the tomatoes can adversely affect segmentation outcomes [3].

Over the past ten years or more, significant research and practical efforts have focussed on detecting and segmenting target fruits. Conventional techniques involve analyzing single features, such as color, geometric shape, and texture. For instance, Si et al. [4] segmented apples from background images using color surface variation analysis. Methods of multi-feature fusion analysis employ either a combination of geometric shape and color attributes or a fusion of color, intensity, edge, and orientation characteristics. Yin et al. [5] designed an approach to recognize ripe tomatoes by initially reducing noise in cases of occlusion and overlap of the fruit and subsequently combining their color and shape attributes for recognition. Nonetheless, both single-feature analysis and multi-feature fusion techniques are compromised by low robustness and extensive time consumption. Detection and segmentation accuracy can be negatively impacted by changes in

environmental conditions and variations in fruit surface color and features, especially in unstructured environments.

The accurate, efficient, and real-time instance segmentation of growing and diseased tomatoes is essential in the complex environment of tomato greenhouses. This enables the timely picking of ripe fruits, helps avoid spoilage, and assists in monitoring diseased fruits to prevent bacterial infections in the planting field. In the past several years, deep learning technology has widely been employed for tasks such as instance segmentation and object detection, owing to its high accuracy and efficiency. To achieve object detection and instance segmentation, a branch for generating binary masks has been introduced to Mask RCNN [6], serving as a prototype of Fast RCNN (Faster R-CNN: Toward real-time object detection using region proposal networks) [7]. Jia et al. [8] used the improved Mask RCNN algorithm for instance segmentation on overlapping apples. They fused ResNet [9] and DenseNet as the feature extraction network of the model and achieved an accuracy rate of 97.31% on 120 images in the test set. Huang et al. [10] proposed a fuzzy Mask R-CNN to automatically identify the maturity of tomato fruits. They distinguished the foreground and background in the image through the fuzzy c-means model and Hough transform method, located the edge features for automatic labeling, and achieved a 98.00% accuracy rate in 100 images. Afonso et al. [11] used the Mask R-CNN model, with ResNet101 as the backbone, to segment ripe and unripe tomatoes, achieving a segmentation precision of 95% and 94% for each, respectively. Wang et al. [12] proposed an improved Mask RCNN model that integrates the attention mechanism for segmenting apple maturity under various conditions, such as light influence, occlusion, and overlap. The test results showed accuracy and recall rates of 95.8% and 97.1%, respectively. In addition, the segmentation of tomato fruit [13], the detection of tomato fruit infection areas [14], the segmentation of tomato maturity [15], and the segmentation of Soil block [16] based on Mask RCNN have demonstrated that the high precision and robustness of the Mask RCNN algorithm in object detection and instance segmentation. Mask RCNN is a conventional two-stage instance segmentation model. Masks are generated by Mask RCNN through feature positioning. The located features are then passed to the mask predictor after performing pooling operations on the region of interest. However, executing these operations sequentially can cause slow segmentation speed, large model size, and an increased number of computing parameters. In contrast to conventional instance segmentation algorithms, which rely on feature localization to generate masks, YOLACT (You Only Look At Coefficients) [17] is a real-time method. It can rapidly generate high-quality instance masks by parallelizing the tasks of generating prototype masks and predicting mask coefficients. The task of instance segmentation, which uses the YOLO framework, builds on the principles of the YOLACT network for completion. Initially, two parallel sub-tasks are executed: generating prototype masks and predicting mask coefficients. Subsequently, the prototype is subjected to linear weighting based on the obtained mask coefficients, which leads to the creation of instance masks. Mubashiru [18] proposed a lightweight YOLOv5 algorithm for accurately segmenting fruits from four gourd family plants with similar features. The proposed algorithm achieved a segmentation accuracy of 88.5%. Although this method attains faster segmentation speed, there is a need to further optimize the accuracy of the segmentation. In contrast to the anchor-based detection head of YOLOv5, YOLOv8 adopts a novel anchor-free method. This method decreases the number of hyperparameters, which improves the model's scalability while enhancing segmentation performance.

This paper proposed an improved YOLOv8s-Seg algorithm for segmenting healthy and diseased tomatoes based on research conducted by scholars worldwide. The research consisted of the following tasks:

(1)　To enhance the edge features of the tomatoes, algorithms such as Gaussian blur, Sobel operator, and weighted superposition were used to sharpen the 1600 photos in the original dataset. Further data enhancement operations expanded the dataset to 9600 photos;

(2) The feature fusion capability of the algorithm was improved by adding SimConv convolution [19] before the two upsampling operations in the feature fusion network, replacing the remaining regular convolutions with SimConv convolution, and swapping the C2f module with the RepBlock module [20];

(3) An improved YOLOv8s-Seg algorithm was proposed to address the slow running time, high parameter count, and large number of calculations of the two-stage instance segmentation model. This algorithm was designed with the aim of effective, real-time instance segmentation of healthy and diseased tomatoes.

## 2. Materials and Methods

### 2.1. Data Acquisition

The dataset includes photographs of tomatoes at four stages of maturity, including young fruit, immature, half-ripe, and ripe. It also includes images of six common tomato diseases: grey mold, umbilical rot, crack, bacterial canker, late blight, and virus disease. A total of 788 photos were captured from tomato cultivation plots 26 and 29 at Shenyang Agricultural University (latitude: 41.8° N), with a seedling spacing of 0.3 m. Furthermore, an additional 812 photos which demonstrate the previously mentioned five diseases, namely grey mold, umbilical rot, bacterial canker, late blight, and virus disease, were retrieved from Wikipedia. This brings the total number of photos in the dataset to 1600. The images were taken using an iPhone 13, which captured them in JPG format with a resolution of 1280 × 720 pixels. Images retrieved from Wikipedia were preserved in the same format and resolution. The dataset was divided into training and validation sets at a 7:3 ratio. As a result, there were 1120 photos in the training set and 480 photos in the validation set. Example images are shown in Figure 1.

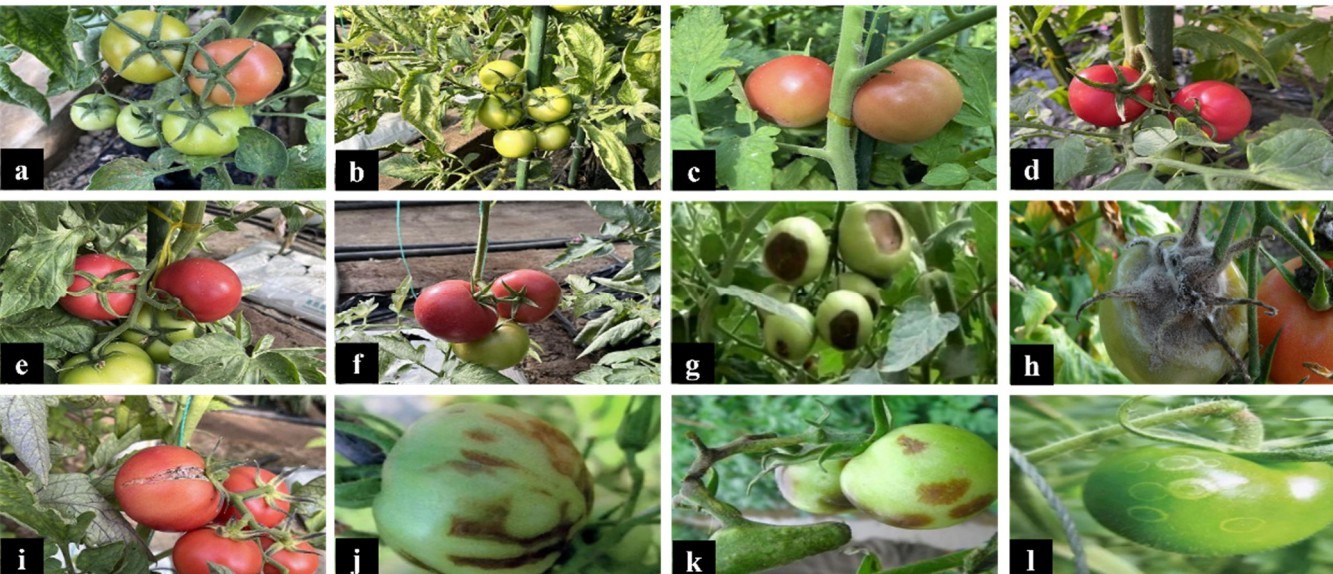

**Figure 1.** Example images. (**a**) young fruit, (**b**) immature, (**c**) half-ripe, (**d**) ripe, (**e,f**) ripe, immature, (**g**) umbilical rot, (**h**) grey mold, (**i**) crack, (**j**) virus disease, (**k**) late blight, (**l**) bacterial canker.

### 2.2. Image Preprocessing

In this study, the dataset of 1600 photos was enhanced using several techniques, including the Sobel operator and weighted overlay. These methods aimed to improve the clarity of tomato fruit edges for better image annotation and feature extraction. Initially, Gaussian blur was applied to the images to reduce noise. Then, the Sobel operator was utilized to calculate the image gradients and extract edge features. Finally, the gradient images were combined with the original images using a weighted overlay technique to enhance the edge features. Figure 2 compares the photos before and after the image sharpening process.

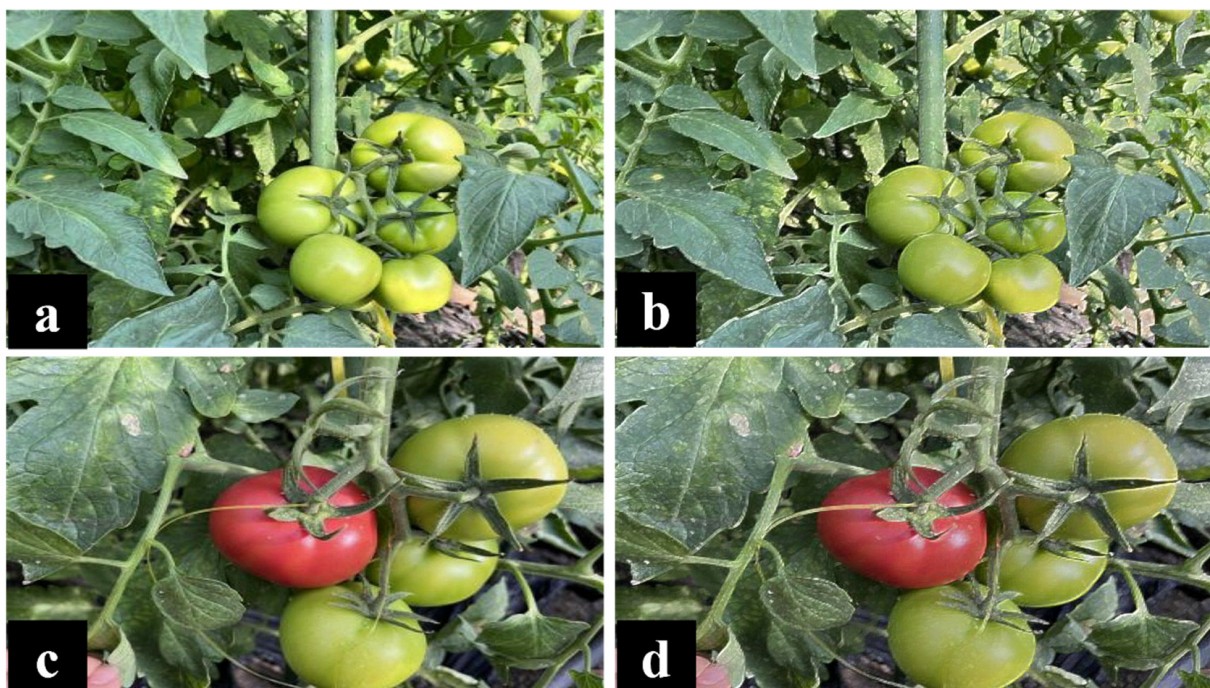

**Figure 2.** Image sharpening. (**a**,**c**) Original image. (**b**,**d**) Sharpened image.

Increasing the number of images through data enhancement can help minimize overfitting during the training process and improve the robustness of the model, thereby increasing the generalization ability of the model. Lighting conditions and shooting angles can significantly influence fruit detection and segmentation in tomato greenhouses. Brightness adjustment, mirroring, and rotation operations were applied to simulate weather changes and variations in detection equipment angles [21]. Performing data enhancement on the sharpened dataset of 1600 photos resulted in the expansion of the dataset to 9600 photos through brightness adjustment, mirroring, and rotation, as shown in Figure 3. The training set includes 6720 photos, and the validation set contains 2880 photos, resulting in 6744 tomato fruits. Table 1 presents the detailed structure.

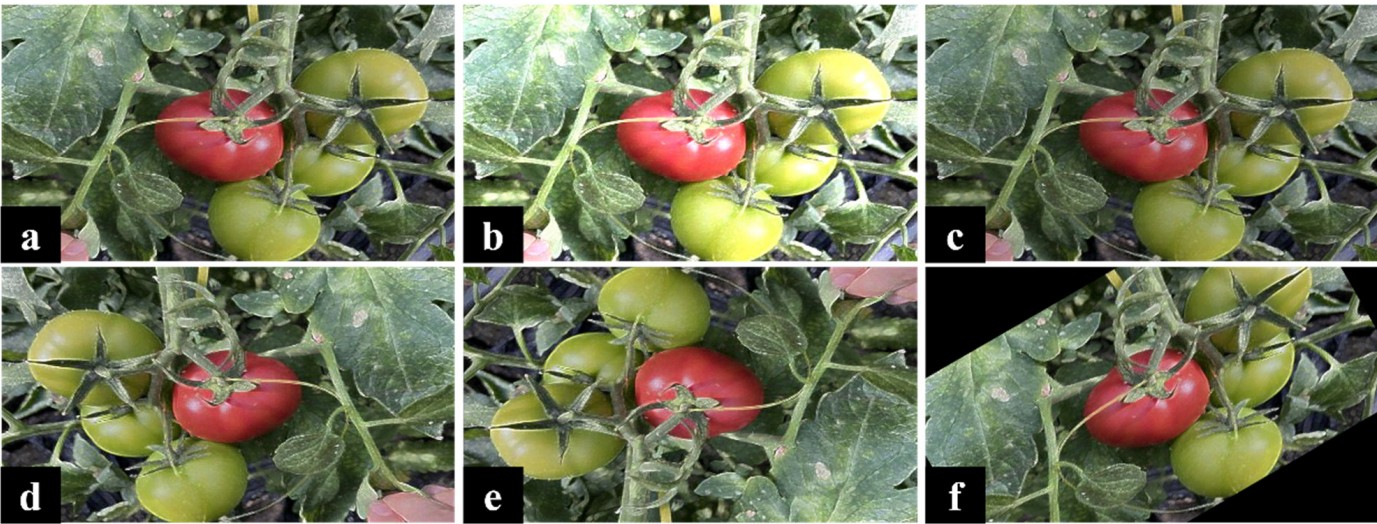

**Figure 3.** Data enhancement. (**a**) sharpened image; (**b**) sharpened image adjusted brightness (brightened); (**c**) sharpened image adjusted brightness (darkened); (**d**) sharpened image subjected to mirroring operation; (**e**) sharpened image rotated by 180°; (**f**) sharpened image rotated by 30°.

**Table 1.** Structure of the dataset.

| Type | Number of Images after Sharpening | Train (Data Enhancement) | Validation (Data Enhancement) | Number of Instances (Validation) |
|------|-----------------------------------|--------------------------|-------------------------------|----------------------------------|
| late blight | 156 | 655 | 281 | 602 |
| crack | 150 | 630 | 270 | 740 |
| grey mold | 152 | 638 | 274 | 593 |
| virus | 164 | 689 | 295 | 589 |
| rot | 174 | 731 | 313 | 601 |
| canker | 166 | 697 | 299 | 579 |
| ripe | 161 | 677 | 289 | 852 |
| half-ripe | 152 | 638 | 274 | 778 |
| immature | 168 | 706 | 302 | 900 |
| young | 157 | 659 | 283 | 780 |
| Total | 1600 | 6720 | 2880 | 6744 |

Late blight indicates tomato late blight; crack indicates tomato crack; grey mold indicates tomato grey mold; virus indicates tomato virus disease; rot indicates tomato umbilical rot; canker indicates tomato bacterial canker; ripe indicates ripe tomato; half-ripe indicates half-ripe tomato; immature indicates immature tomato; and young indicates young fruit tomato.

Ten categories of photos were manually annotated using Labelme software (version 6.1.1) after completing the data enhancement. These categories are tomato late blight, tomato crack, tomato grey mold, tomato virus diseases, tomato umbilical rot, tomato bacterial canker, ripe tomatoes, half-ripe tomatoes, immature tomatoes, and young fruit. During the annotation process, the polygon tool was selected to annotate the edges of the tomatoes that required instance segmentation. Figure 4 displays the annotation of tomatoes.

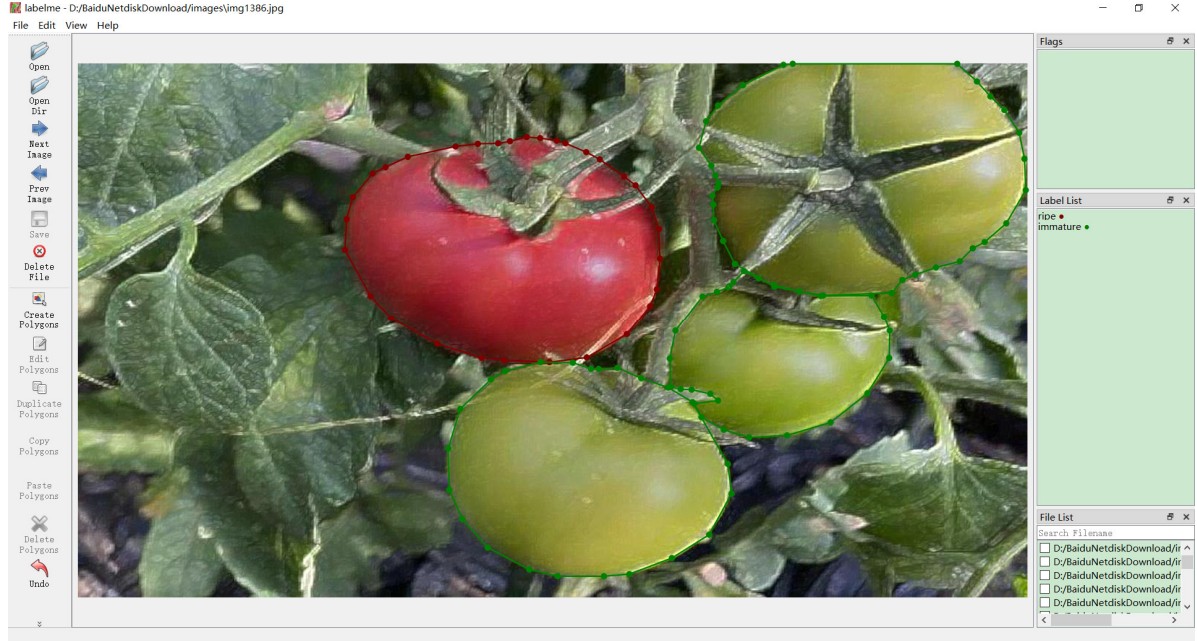

**Figure 4.** Annotation of tomatoes.

*2.3. Tomato Instance Segmentation Based on Improved YOLOv8s-Seg*

The YOLO (you only look once) series is a deep-learning model for detecting objects. YOLOv8, developed by the same authors as YOLOv5, shares a similar overall style. YOLOv8 has made significant improvements and optimizations over the YOLOv5 network, resulting in enhanced algorithm performance. The YOLOv8 network supports object detection and tracking, as well as additional tasks, such as instance segmentation, image classification, and key point detection. Similar to YOLOv5, YOLOv8 provides five different scales of models (n, s, m, l, x), with increasing depth and width from left to right. In reference to the ELAN design philosophy [22], YOLOv8 replaces the C3 structure in

the YOLOv5 backbone network with a C2f structure. This alteration enables YOLOv8 to maintain its lightweight characteristics while obtaining a greater amount of gradient flow information. Compared to YOLOv5, the head part of YOLOv8 exhibits more prominent differences due to the implementation of the widely-used decoupled head structure. For loss function calculation, YOLOv8 utilizes the TaskAlignedAssigner positive sample assignment strategy [23]. Furthermore, it introduces the distribution focal loss [24]. During training, the strategy of disabling mosaic augmentation in the last 10 epochs is incorporated, as introduced in YOLOX [25], to effectively improve precision in the data augmentation process. YOLOv8s-Seg is an extension of the YOLOv8 object detection model. It is specifically designed for carrying out segmentation tasks. The YOLOv8s-Seg network draws on the principles of the YOLACT network to achieve real-time instance segmentation of objects and maintain a high segment mean average precision. Figure 5 displays the Structure of the YOLACT network.

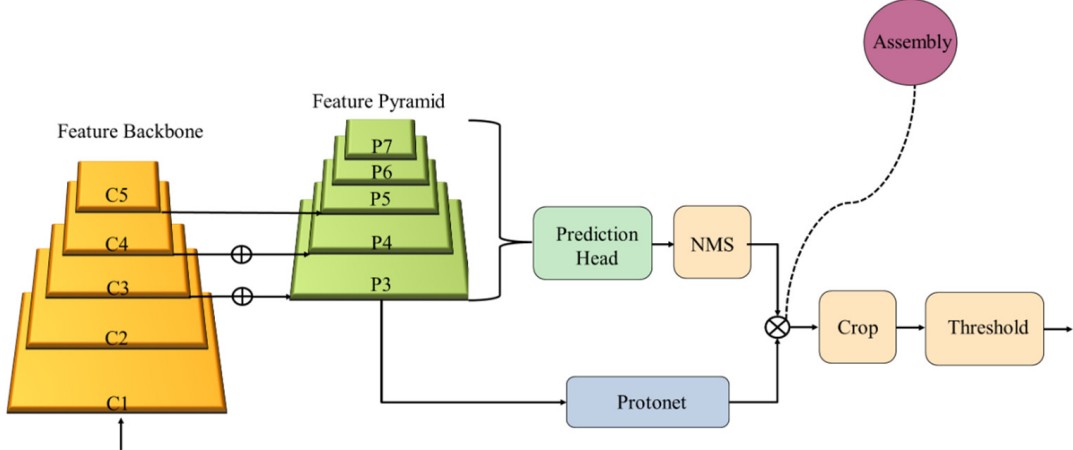

**Figure 5.** Structure of YOLACT network.

The YOLOv8-Seg (ultralytics-8.0.57) network consists of two main components: backbone and head (which can be further divided into neck and segment). The GitHub provides five different scale models of the network, namely, YOLOv8n-Seg, YOLOv8s-Seg, YOLOv8m-Seg, YOLOv8l-Seg, and YOLOv8x-Seg. In this study, experiments were conducted on YOLOv8-Seg models of different scales to evaluate the segment $\text{mAP}_{@0.5}$ and model size. Table 2 presents the results.

**Table 2.** Comparison of network segmentation results.

| Models | Seg $\text{mAP}_{@0.5}$ (%) | Model Size (MB) |
| --- | --- | --- |
| YOLOv8n-Seg | 0.853 | 6.5 |
| YOLOv8s-Seg | 0.898 | 20.4 |
| YOLOv8m-Seg | 0.900 | 54.8 |
| YOLOv8l-Seg | 0.903 | 92.3 |
| YOLOv8x-Seg | 0.907 | 143.9 |

Table 2 shows that YOLOv8s-Seg achieved a segment $\text{mAP}_{@0.5}$ of 89.8%, a 3.5% improvement over YOLOv8n-Seg. However, it was slightly lower than YOLOv8m-Seg, YOLOv8l-Seg, and YOLOv8x-Seg by 0.2%, 0.5%, and 0.9%, respectively. Regarding model size, YOLOv8s-Seg occupies 20.4 MB, an increase of 13.9 MB compared to YOLOv8n-Seg. However, it is significantly lighter than YOLOv8m-Seg, YOLOv8l-Seg, and YOLOv8x-Seg, with reductions of 34.4 MB, 71.9 MB, and 123.5 MB, respectively. Considering the segment $\text{mAP}_{@0.5}$ performance and lightweight requirements, YOLOv8s-Seg was selected as the model for experimentation in this study. Figure 6 illustrates the structure of the improved YOLOv8s-Seg network.

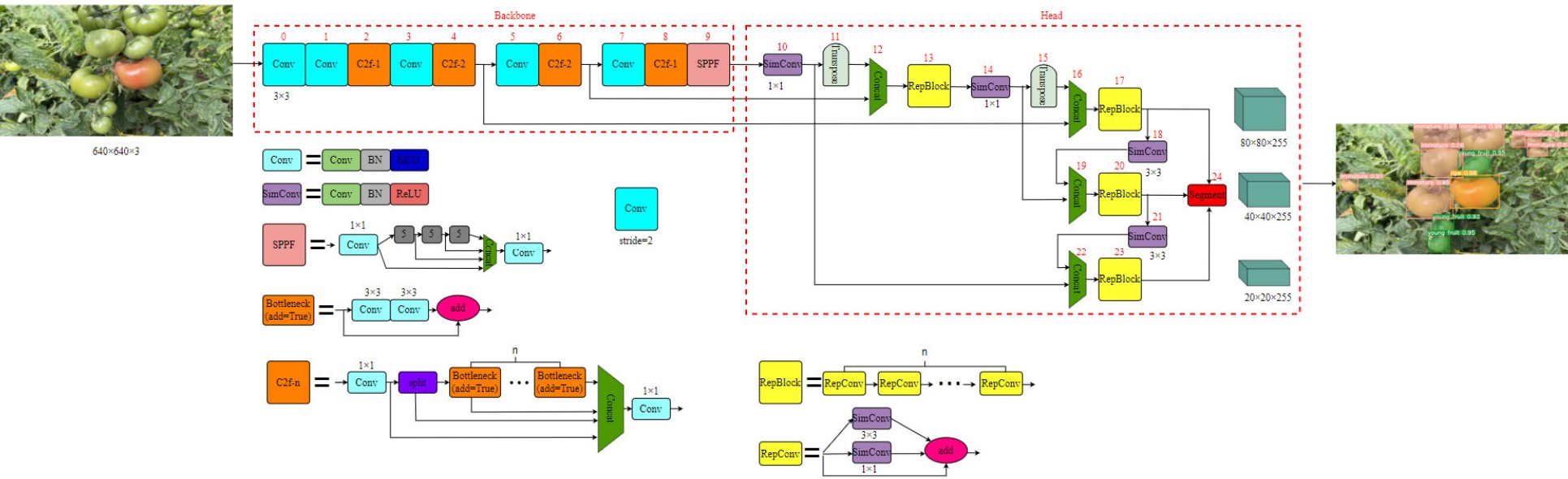

**Figure 6.** Structure of tomato segmentation network based on improved YOLOv8s-Seg. The ability to fuse features of the network was improved by replacing the C2f module with the RepBlock module, adding SimConv convolution before two upsampling in the neck module, and replacing the remaining conventional convolution with SimConv.

The backbone network of YOLOv8s-Seg consists of a $3 \times 3$ convolution, a C2f module, and an SPPF (spatial pyramid pooling fusion) module. In contrast to the YOLOv5 network, YOLOv8s-Seg replaces the initial $6 \times 6$ convolution with a $3 \times 3$ convolution in the backbone network, making the model more lightweight. Additionally, the C3 module (Figure 7) in YOLOv5 is replaced with the C2f module in YOLOv8s-Seg. The C2f module, designed with skip connections and additional split operations, enriches the gradient flow during backpropagation and improves the performance of the model. YOLOv8s-Seg utilizes two versions of the cross stage partial network (CSP). The CSP [26] in the backbone network employs residual connections (as shown in Figure 6), while the head part uses direct connections. The SPPF structure in YOLOv8s-Seg remains the same as in YOLOv5 (version 6.1), utilizing cascaded $5 \times 5$ pooling kernels to accelerate network operation speed.

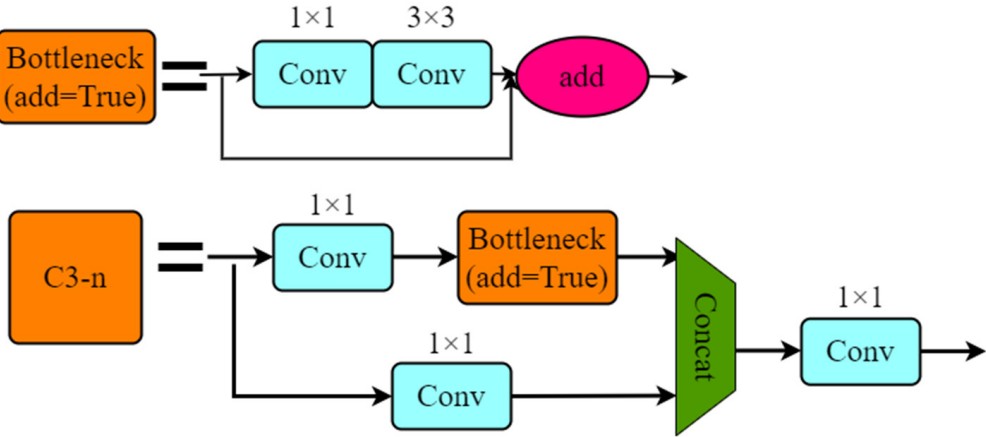

**Figure 7.** Structure of the C3 module.

The head module is comprised of the neck and segment parts. The neck module incorporates the path aggregation network (PANet) [27] and feature pyramid network (FPN) [28] as feature fusion networks. Unlike YOLOv5 and YOLOv6, YOLOv8s-Seg removes the $1 \times 1$ convolution before upsampling and fuses the feature maps directly from different stages of the backbone network. This study aimed to enhance the network performance of YOLOv8s-Seg by improving its neck module. Specifically, before each upsampling operation, two $1 \times 1$ SimConv convolutions were added, and the remaining regular convolutions in the neck part were replaced with $3 \times 3$ SimConv convolutions. The C2f module (Figure 8) was replaced with the RepBlock module (Figure 6). The RepBlock module is composed of stacked RepConv convolutions, and the structure of the RepConv convolution is depicted in Figure 6.

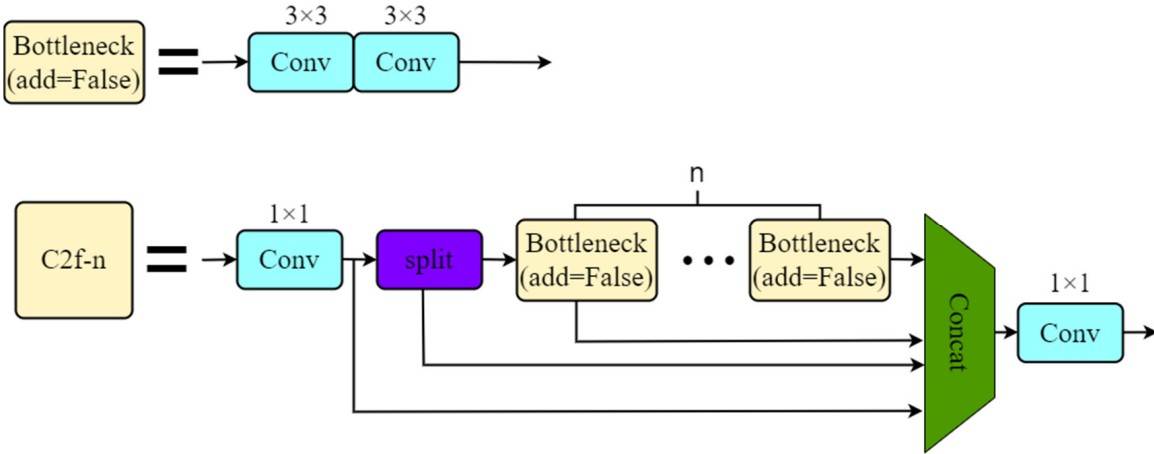

**Figure 8.** Structure of the C2f module in the neck.

The YOLOv5 network employs a static allocation strategy to assign positive and negative samples based on the intersection over union (IOU) between the predicted boxes and ground truth. However, the YOLOv8s-Seg network has improved this aspect by introducing a superior dynamic allocation strategy. It incorporates the TaskAlignedAssigner (TOOD), which selects positive samples based on a weighted score that comes from the classification and regression scores. The computation is represented by Formula (1).

$$t = s^{\alpha} \times u^{\beta} \tag{1}$$

where $s$: prediction scores for labeled categories, $u$: prediction frame with the IOU of Ground Truth, $t$: alignment scores for categorical regression.

During training, YOLOv8s-Seg performs online image enhancement to ensure that the model encounters slightly different images in each epoch. Mosaic enhancement is a crucial method of data improvement that randomly combines four images. This technique compels the model to learn how to detect partially obstructed and differently positioned objects. In the last 10 training epochs, the YOLOv8s-Seg network deactivates the mosaic enhancement, a method proven to improve network precision effectively. Figure 9 demonstrates an example of mosaic enhancement.

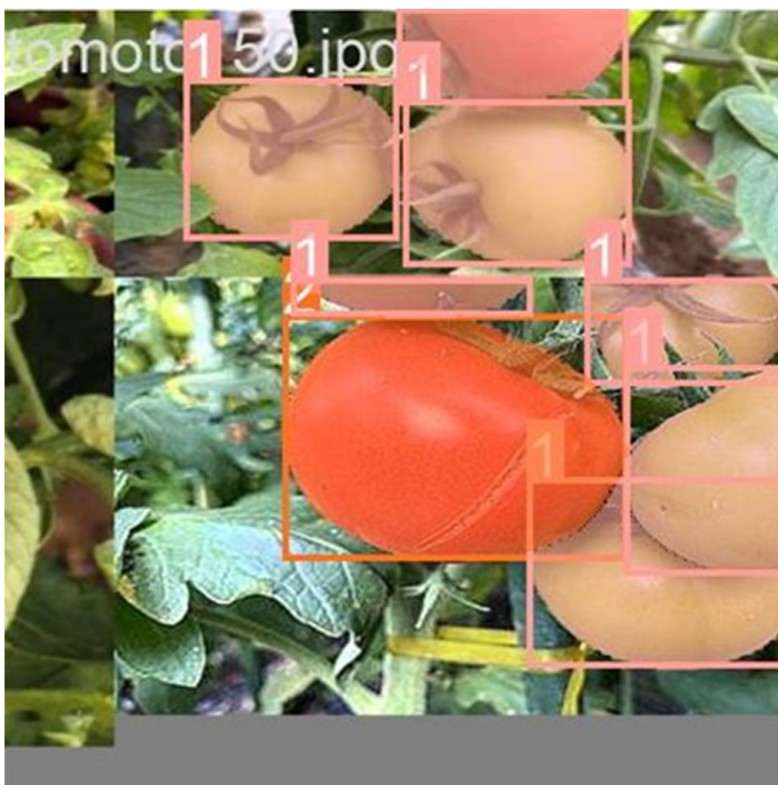

**Figure 9.** Mosaic data enhancement.

*2.4. Model Training and Performance Evaluation*

The examinations were performed on Windows 10 using a 12 vCPU Intel(R) Xeon(R) Platinum 8255C CPU @2.50 GHz and an Nvidia GeForce RTX 2080Ti graphics card. The framework for deep learning was PyTorch 1.8.1 and Compute Unified DeviceArchitecture (CUDA) 11.1, accelerated by cuDNN version 8.0.5. In this experiment, the improved YOLOv8s-Seg, YOLOv8s-Seg, YOLOv5s-Seg, and YOLOv7-Seg were performed in the same environment configuration and under the same hyperparameter settings, as indicated in Table 3. The hyperparameter settings of Mask CNN: the learning rate, batch size, learning momentum, weight decay, number of iterations, and image size were set to 0.004, 2, 0.9, $1 \times 10^{-4}$, 30 epochs, and $640 \times 640$ pixels, respectively.

**Table 3.** Hyperparameters during training.

| | |
|---|---|
| learning rate | 0.01 |
| batch size | 16 |
| momentum | 0.937 |
| weight decay | 0.0005 |
| number of iterations | 300 epochs |
| image size | $640 \times 640$ pixels |

In this study, we assess the performance of the improved YOLOv8s-Seg using precision, recall, F1 score, and segment mAP$_{@0.5}$. Tomato locations were assessed using precision, recall, and F1 score, while segmentation results were evaluated using segment mAP [29]. Equations (2)–(5) are used to calculate the precision, recall, F1 score, and segment mAP scores. The higher the four parameters are, the better the segmentation results.

$$\text{precision} = \frac{\text{TP}}{(\text{TP} + \text{FP})} \times 100\% \tag{2}$$

$$\text{recall} = \frac{\text{TP}}{(\text{TP} + \text{FN})} \times 100\% \tag{3}$$

$$\text{F1} = 2 \times \text{precision} \times \frac{\text{recall}}{\text{precision} + \text{recall}} \tag{4}$$

$$\text{seg}_{\text{mAP}} = \sum_{i=1}^{c} \frac{\text{AP}(i)}{\text{C}} \tag{5}$$

where TP denotes an actual positive sample with a positive prediction, while FP indicates an actual negative sample with a positive prediction, and FN indicates an actual positive sample with a negative prediction. AP represents the average precision of segmentation. The segmentation performance of the model increases with the AP score. C represents the number of segmentation categories.

## 3. Results and Discussion

### 3.1. Instance Segmentation between Growing and Diseased Tomatoes

To validate the performance of the improved YOLOv8s-Seg in segmenting tomato growth stages and common diseases in fruits, we used 2880 photos containing 6744 tomato fruits for validation. The experimental results showed that the model achieved the precision, recall, F1 score, and segment mAP$_{@0.5}$ of 91.9%, 85.8%, 89.0%, and 92.2%, respectively. Figure 10 presents examples of instance segmentation on tomatoes affected by several factors like leaf occlusion, fruit overlap, lighting variations, angle changes, growth stages, and common diseases of the improved YOLOv8s-Seg network. Table 4 shows the results of the segmentation for growing and diseased tomatoes. As shown in Figure 10, the improved YOLOv8s-Seg algorithm accurately segments tomatoes (Figure 10d–f,j) affected by leaf occlusion (Figure 10a), fruit overlap (Figure 10b), lighting variations (Figure 10c), and angle changes (Figure 10g). Overall, the algorithm exhibits precise segmentation performance on tomatoes affected by factors such as leaf occlusion, fruit overlap, lighting variations, and angle changes. Meanwhile, the algorithm also achieves outstanding instance segmentation results for tomatoes (Figure 10k,l) in growth stages (Figure 10h) and those affected by disease (Figure 10i). From Table 4, it can be seen that the precision rates obtained by the improved YOLOv8s-Seg algorithm for instance segmentation of ripe tomato, half-ripe tomato, immature tomato, young fruit, grey mold, umbilical rot, bacterial canker, late blight, virus disease, and crack scores were 92.7%, 92.3%, 89.9%, 91.2%, 92.6%, 92.2%, 91.5%, 92.4%, 93%, and 91.3% respectively. The analysis of instance segmentation results on tomatoes in growth stages and diseased tomatoes reveals the effectiveness of the algorithm in overcoming the impact of tomato surface color and features on segmentation performance.

In conclusion, the algorithm exhibits remarkable segmentation performance on tomatoes affected by environmental factors during growth stages and disease.

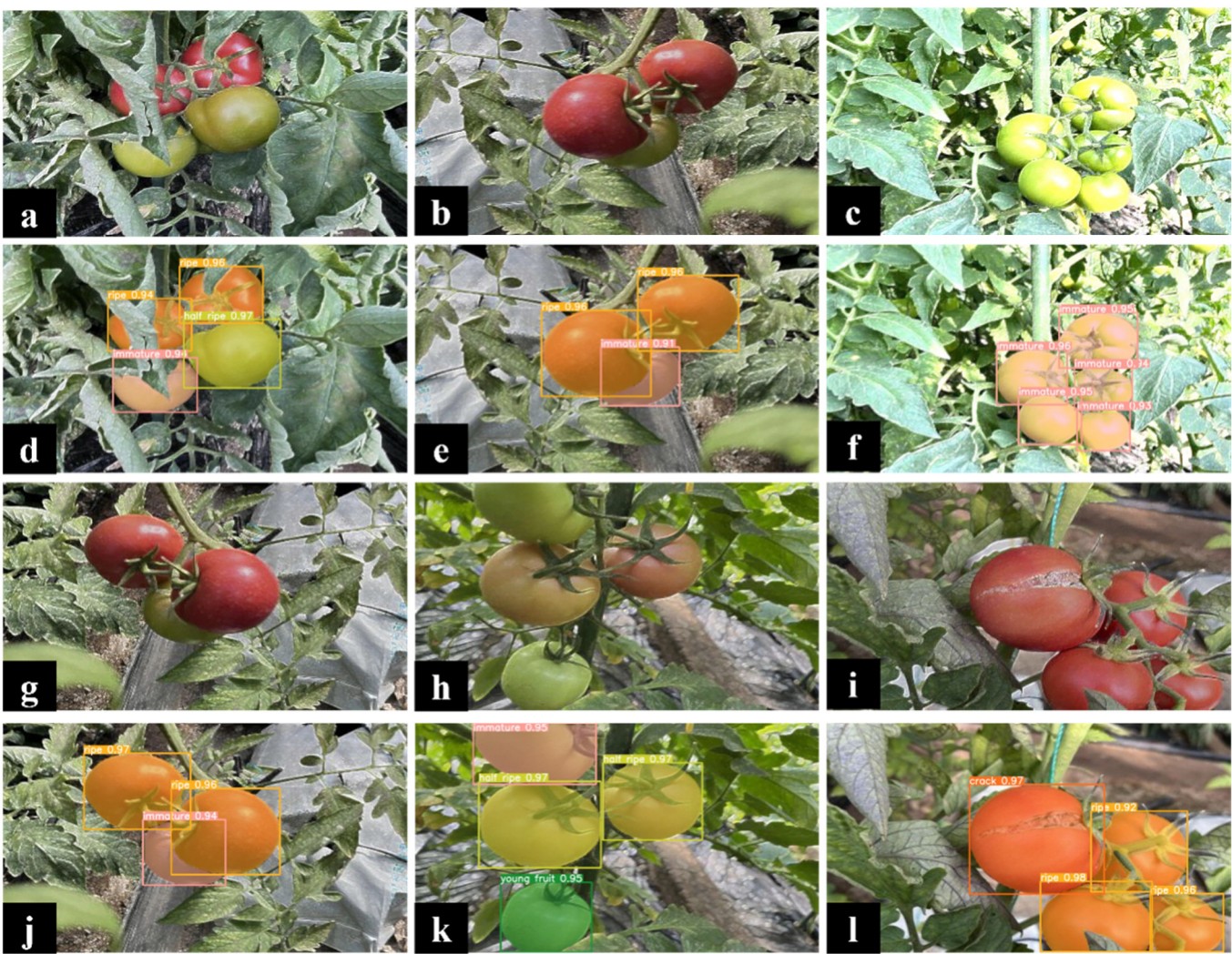

**Figure 10.** Examples of instance segmentation of tomatoes. (**a**): ripe tomatoes and immature tomatoes shaded by leaves, half-ripe tomatoes with intact fruit characteristics; (**b**): immature tomatoes with overlapping fruit and ripe tomatoes with intact fruit characteristics; (**c**): immature tomatoes affected by changes in light; (**d**): example segmentation of tomatoes shaded by leaves; (**e**): example segmentation of overlapping fruit; (**f**): example segmentation of tomatoes affected by changes in light; (**g**): immature and ripe tomatoes affected by changes in angle; (**h**): immature tomatoes, half-ripe tomatoes, and young fruit; (**i**): cracked tomatoes, ripe tomatoes. (**j**): example segmentation of tomatoes affected by changes in angle; (**k**): example segmentation for immature tomatoes, half-ripe tomatoes, and young fruit; (**l**): example segmentation for cracked tomatoes and ripe tomatoes.

**Table 4.** Segmentation results of healthy and diseased tomatoes.

| Type | Canker | Immature | Crack | Ripe | Half-Ripe | Grey Mold | Late Blight | Rot | Young Fruit | Virus |
|------|--------|----------|-------|------|-----------|-----------|-------------|-----|-------------|-------|
| precision (%) | 91.5 | 89.9 | 91.3 | 92.7 | 92.3 | 92.6 | 92.4 | 92.2 | 91.2 | 93 |

Late blight indicates tomato late blight; crack indicates tomato crack; grey mold indicates tomato grey mold; virus indicates tomato virus disease; rot indicates tomato umbilical rot; canker indicates tomato bacterial canker; ripe indicates ripe tomato; half-ripe indicates half-ripe tomato; immature indicates immature tomato; and young indicates young fruit tomato.

### 3.2. Comparison with Other Instance Segmentation Algorithms

In this paper, to investigate the segmentation capabilities of the improved YOLOv8s-Seg for tomatoes, the performance of the network was evaluated in terms of precision, recall, F1 score, segment mAP$_{@0.5}$, and inference time. This performance was then compared with that of YOLOv8s-Seg, YOLOv7-Seg, YOLOv5s-Seg, and Mask RCNN algorithms. The example segmentation results of the five networks were obtained using the same training and validation sets during the training process. The hyperparameters of the five models during the training process were set as follows: the hyperparameters of YOLOv8s-Seg, the improved YOLOv8s-Seg, the YOLOv5s-Seg, and YOLOv7-Seg are shown in Table 3. For Mask RCNN, the learning rate, batch size, learning momentum, weight decay, and number of iterations were set to 0.004, 2, 0.9, $1 \times 10^{-4}$, and 30 epochs, respectively. Table 5 displays the segmentation results for the five models.

**Table 5.** Segmentation results for the five algorithms.

| Method | Precision (%) | Recall (%) | F1 Score (%) | Segment mAP$_{@0.5}$ | Inference Time (ms) |
|---|---|---|---|---|---|
| Mask RCNN | 89.8 | 85.5 | 87.6 | 0.915 | 90 |
| YOLOv5s-Seg | 89.0 | 83.0 | 85.9 | 0.890 | 2.5 |
| YOLOv7-Seg | 91.4 | 84.8 | 87.9 | 0.904 | 15.2 |
| YOLOv8s-Seg | 90.3 | 85.4 | 87.7 | 0.898 | 3.1 |
| Improved YOLOv8s-Seg(ours) | 91.9 | 85.8 | 88.7 | 0.922 | 3.5 |

In Table 5, the results show the performance of the improved YOLOv8s-Seg algorithm compared to other models. The improved YOLOv8s-Seg algorithm achieves precision, recall, F1 score, and segment mAP$_{@0.5}$ of 91.9%, 85.8%, 88.7%, and 0.922, respectively. Compared to the YOLOv8s-Seg algorithm, the improvements were 1.6%, 0.4%, 1.0%, and 2.4%, respectively. Compared to the YOLOv5s-Seg algorithm, the improvements were 2.9%, 2.8%, 2.8%, and 3.2%, respectively. Compared to the YOLOv7-Seg algorithm, this algorithm showed increases of 0.5%, 1.0%, 0.8%, and 1.8%. Compared to the Mask RCNN algorithm, this algorithm had increments of 2.1%, 0.3%, 1.1%, and 0.7%, respectively. Additionally, the inference time of 3.5 ms signifies a minor increase over YOLOv5s-Seg and YOLOv8s-Seg (0.4 ms and 0.6 ms) but a significant reduction over YOLOv7-Seg and Mask RCNN (11.7 ms and 86.5 ms), supporting real-time instance segmentation. In conclusion, the improved YOLOv8s-Seg algorithm stands out in precision, recall, F1 score, and segment mAP$_{@0.5}$, with effective inference time. Figure 11 provides the comparison of Segment mAP$_{@0.5}$ for five algorithms.

### 3.3. Comparison of the Improved YOLOv8s-Seg and YOLOv8s-Seg

In this paper, we proposed an improved YOLOv8s-Seg network designed for real-time and effective instance segmentation of various tomato stages, including young fruit, immature, half-ripe, ripe, and common diseases such as grey mold, umbilical rot, crack, bacterial canker, late blight, and virus disease. The feature fusion capability of the algorithm has been significantly improved through enhancements to the feature fusion network, with modifications detailed in Table 6. Analysis of Tables 5 and 6 shows that the improved YOLOv8s-Seg network achieved an F1 score of 88.7% and a segment mAP$_{@0.5}$ of 92.2%, representing improvements of 1.0% and 2.4%, respectively, over the original YOLOv8s-Seg network. Notably, the segment mAP$_{@0.5}$ improvements were achieved with only a marginal increase in memory size (0.7 MB) and inference time (0.4 ms), highlighting the network's efficiency in real-time instance segmentation of both growing and diseased tomatoes.

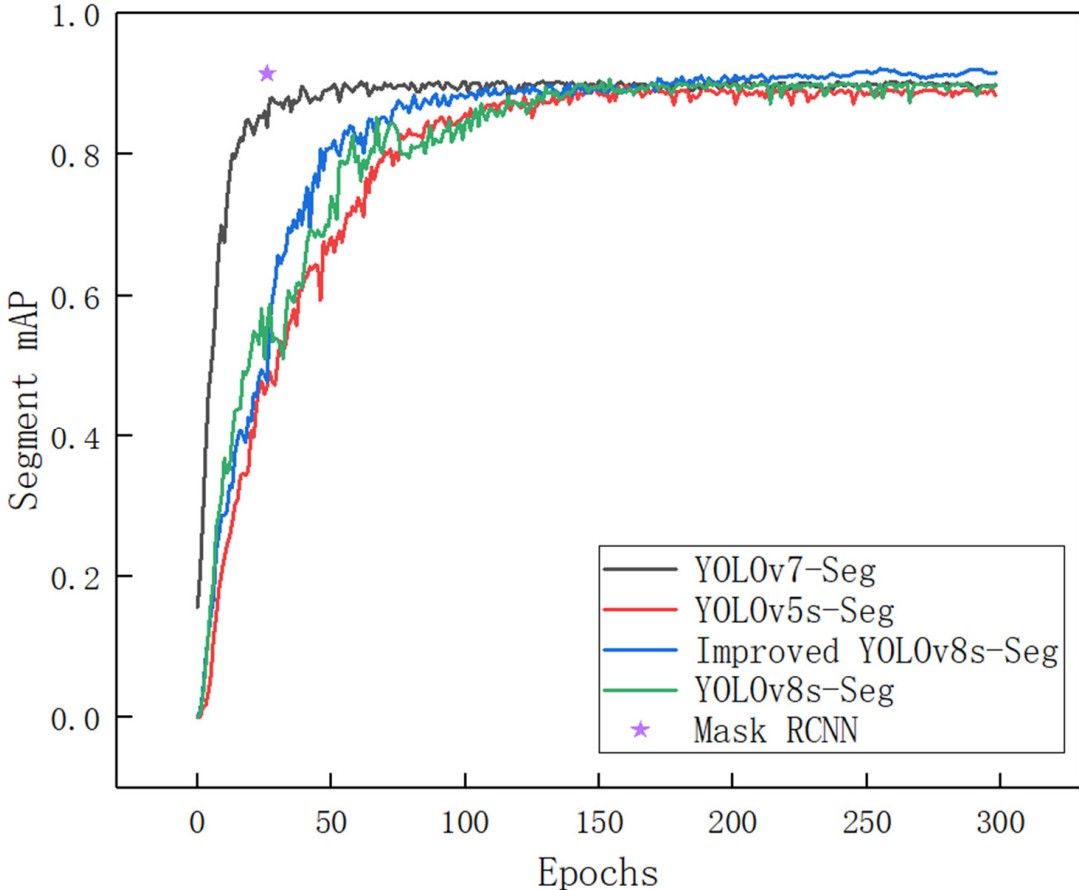

**Figure 11.** Comparison of Segment mAP$_{@0.5}$ for five algorithms.

**Table 6.** Comparison of parameter variations.

| Methods | Model Size | Δ% MB | GFLOPs | Δ% FLOPs | Parameters | Δ% Parameters | Inference Time (ms) |
|---|---|---|---|---|---|---|---|
| YOLOv8s-Seg | 20.4 | | 42.5 | | 11,783,470 | | 3.1 |
| Improved YOLOv8s-Seg | 21.1 | +0.7 | 47.2 | +4.7 | 10,400,750 | −1,382,720 | 3.5 |

*3.4. Effect of Different Image Resolutions on Tomato Segmentation*

To investigate the impact of photo resolution on the segmentation results of the improved YOLOv8s-Seg network, we experimented using different input image sizes during training: 416 × 416 pixels, 640 × 640 pixels, 768 × 768 pixels, and 1024 × 1024 pixels. Table 7 provides a comprehensive overview of the segment mAP$_{@0.5}$ and the inference time for each image resolution. The results reveal that as the photo resolution increases from 416 × 416 pixels to 640 × 640 pixels, the inference time increases by 2.6 ms, while the segment mAP$_{@0.5}$ improves by 1.1%. This indicates that the model enhances the instance segmentation performance at the cost of a slight increase in inference time. However, when the resolution is further increased to 768 × 768 pixels and 1024 × 1024 pixels, the inference time shows more substantial increments of 4.4 ms and 6.3 ms, respectively. In contrast, the segment mAP$_{@0.5}$ only experiences minor improvements of 0.2% and 0.3%. It could be concluded that the resolution of 640 × 640 pixels is more suitable for training the improved YOLOv8s-Seg network. This resolution balances achieving satisfactory segmentation performance and maintaining reasonable inference time.

**Table 7.** Comparison of network segmentation results.

| Resolutions (Pixels) | Segment mAP$_{@0.5}$ (%) | Inference Time (ms) |
| --- | --- | --- |
| 416 × 416 pixels | 91.1 | 0.9 |
| 640 × 640 pixels | 92.2 | 3.5 |
| 768 × 768 pixels | 92.4 | 7.9 |
| 1024 × 1024 pixels | 92.5 | 9.8 |

## 4. Conclusions

An improved YOLOv8s-Seg network based on instance segmentation for tomato illness and maturity was suggested in this paper. The feature fusion capability of the algorithm was improved by replacing the C2f module with the RepBlock module, adding SimConv convolution before two upsampling in the feature fusion network, and replacing the remaining conventional convolution with SimConv. The improved YOLOv8s-Seg network achieved a segment mAP$_{@0.5}$ of 92.2% on the validation set. This showed an improvement of 2.4% compared to the original YOLOv8s-Seg network, an improvement of 3.2% over the YOLOv5s-Seg network, an improvement of 1.8% relative to the YOLOv7-Seg network, and an improvement of 0.7% over the Mask RCNN network. Regarding inference time, the improved YOLOv8s-Seg network reached a speed of 3.5 ms, an increase of 0.4 ms and 0.6 ms compared to the YOLOv8s-Seg and YOLOv5s-Seg networks, but a significant reduction compared to the YOLOv7-Seg and Mask RCNN algorithms, reduced by 11.7 ms and 86.5 ms respectively. This capability facilitates the real-time segmentation of both healthy and diseased tomatoes. Overall, the improved YOLOv8s-Seg network exhibits precise segmentation performance on tomatoes affected by factors such as leaf occlusion, fruit overlap, lighting variations, and angle changes. Meanwhile, the analysis of instance segmentation results for tomatoes at different growth stages and diseases shows that the algorithm effectively reduces the impact of surface color and features on performance.

In conclusion, the algorithm shows notable segmentation performance on tomatoes affected by environmental factors during growth stages and disease. Future research will continue to optimize the algorithm to improve the segment mAP$_{@0.5}$. Efforts will also be directed toward simplifying the YOLOv8s-Seg network structure to increase computational efficiency.

**Author Contributions:** Conceptualization, X.Y.; methodology, K.Q.; software, K.Q.; validation, X.N. and Y.Z.; formal analysis, Y.L.; investigation, Y.L.; resources, Y.Z.; data curation, X.N.; writing—original draft preparation, K.Q.; writing—review and editing, X.Y.; visualization, K.Q.; supervision, C.L.; project administration, X.Y.; funding acquisition, X.Y. All authors have read and agreed to the published version of the manuscript.

**Funding:** This work was supported in part by the Youth Program of the Liaoning Education Department under Grant LSNQN202025.

**Institutional Review Board Statement:** Not applicable.

**Data Availability Statement:** Data will be made available on request.

**Conflicts of Interest:** We have no affiliations with any organization with a direct or indirect financial interest in the subject matter discussed in the manuscript.

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
