# Peer review of "Improved YOLOv8-Seg Network for Instance Segmentation of Healthy and Diseased Tomato Plants in the Growth Stage"

_agriculture, doi:10.3390/agriculture13081643_

Round 1
Reviewer 1 Report
The research article titled "Improved YOLOv8-Seg Network for Instance Segmentation of Healthy and Diseased Tomato Plants in the Growth Stage" takes an innovative approach to address the critical issue of crop diseases, particularly in tomato plants. The authors put forth an advanced algorithm, based on the YOLOv8-Seg Network, which has shown promising results in the real-time and efficient segmentation of tomato fruits. The precision, recall, F1 score, and segment mAP@0.5 of the improved algorithm demonstrate marked improvements over the previous models, which is truly commendable. The fact that the inference time of the improved network is only 3.5ms further showcases its potential for practical application in the field of agriculture.
However, to strengthen the scientific contribution and ensure better understanding of the model, I propose the following suggestions:
- Image Aspect Ratio: To assist the readers and visual learners, it would be beneficial if the aspect ratio of the images were modified to ensure they are clear and of good size. This would also help to better visualize the algorithm's performance and the severity of diseases in the segmented images.
- Overfitting Validation: Although the proposed algorithm shows impressive results, more information about how overfitting is prevented during model training would be beneficial. This could include details on cross-validation, early stopping, or dropout methods used, if any.
- Comparison with Existing Models: The current advancement in segmentation science is impressive, with models like Facebook's Segment Anything Model (SAM) making waves. Comparing the YOLOv8-Seg network's performance with such models could add valuable context and more robustly validate the proposed algorithm's efficacy.
- Language Accuracy: While the technical content of the paper is well-structured, there are areas where the English language can be improved. Ensuring language accuracy and clarity can help in conveying your findings more effectively to the readership.
In conclusion, the research presents a significant step forward in using advanced segmentation techniques for monitoring the health of tomato plants and potentially other crops as well. With a few modifications, this study could become an even stronger contribution to the ongoing technological advancements in the agriculture sector.
- Language Accuracy: While the technical content of the paper is well-structured, there are areas where the English language can be improved. Ensuring language accuracy and clarity can help in conveying your findings more effectively to the readership.
Reviewer 2 Report
Authors have proposed an improved YOLOv8s-Seg network to achieve real-time and efficient segmentation of tomato fruits, surface color, and surface features, Authors have developed a robust model and analysed it with suitable parameters. Overall some minor corrections are required.
1. Authors are advised to thoroughly check the references and compare the performance of developed model with published work.
2. It is advised to include the details of hyperparameters tuning. Since dataset is small less than 1000 images for each class after augmentation so hyperparameter tuning is crucial to develop an efficient model with good generalization capability. learning rate of 0.01 and a maximum epoch count of 300 are used in the work. Are these optimized hyperparameters? if yes, what method was used?
3. Authors are advised to thoroughly proofread the paper for better understanding in terms of usage of language. For example, Authors have written " The segmentation effect is stronger, and the model's performance is more dependable and efficient the higher the generated score. “ Here intent of authors is not clear.
4. One more question have you considered the precision and recall by class?
Presentation needs improvement
